# Purification and Identification of Novel Xanthine Oxidase Inhibitory Peptides Derived from Round Scad (*Decapterus maruadsi*) Protein Hydrolysates

**DOI:** 10.3390/md19100538

**Published:** 2021-09-24

**Authors:** Xiao Hu, Ya Zhou, Shaobo Zhou, Shengjun Chen, Yanyan Wu, Laihao Li, Xianqing Yang

**Affiliations:** 1Key Laboratory of Aquatic Product Processing, Ministry of Agriculture and Rural, South China Sea Fisheries Research Institute, Chinese Academy of Fishery Sciences, Guangzhou 510300, China; huxiao@scsfri.ac.cn (X.H.); zhouya_zy@163.com (Y.Z.); chenshengjun@scsfri.ac.cn (S.C.); wyy@scsfri.ac.cn (Y.W.); lilaihao@scsfri.ac.cn (L.L.); 2Co-Innovation Center of Jiangsu Marine Bio-Industry Technology, Jiangsu Ocean University, Lianyungang 222005, China; 3College of Food Science & Technology, Shanghai Ocean University, Shanghai 201306, China; 4School of Life Sciences, Institute of Biomedical and Environmental Science and Technology, University of Bedfordshire, Luton LU1 3JU, UK; shaobo.zhou@beds.ac.uk; 5Collaborative Innovation Center of Provincial and Ministerial Co-Construction for Marine Food Deep Processing, Dalian 116034, China

**Keywords:** round scad (*Decapterus maruadsi*), hydrolysis, peptides, xanthine oxidase inhibitory, purification, identification

## Abstract

The objective of the present study was to investigate the xanthine oxidase (XO) inhibitory effects of peptides purified and identified from round scad (*Decapterus maruadsi*) hydrolysates (RSHs). In this study, RSHs were obtained by using three proteases (neutrase, protamex and alcalase). Among them, the RSHs of 6-h hydrolysis by neutrase displayed the strongest XO inhibitory activity and had an abundance of small peptides (<500 Da). Four novel peptides were purified by immobilized metal affinity chromatography and identified by nano-high-performance liquid chromatography mass/mass spectrometry. Their amino acid sequences were KGFP (447.53 Da), FPSV (448.51 Da), FPFP (506.59 Da) and WPDGR (629.66 Da), respectively. Then the peptides were synthesized to evaluate their XO inhibitory activity. The results indicated that the peptides of both FPSV (5 mM) and FPFP (5 mM) exhibited higher XO inhibitory activity (22.61 ± 1.81% and 20.09 ± 2.41% respectively). Fluorescence spectra assay demonstrated that the fluorescence quenching mechanism of XO by these inhibitors (FPSV and FPFP) was a static quenching procedure. The study of inhibition kinetics suggested that the inhibition of both FPSV and FPFP was reversible, and the type of their inhibition was a mixed one. Molecular docking revealed the importance of π-π stacking between Phe residue (contained in peptides) and Phe^914^ (contained in the XO) in the XO inhibitory activity of the peptides.

## 1. Introduction

Hyperuricemia is currently recognized as the fourth highest-risk chronic disease after hyperglycemia, hypertension, and hyperlipidemia, which could cause complications such as gout, hypertension and diabetes disease [1]. It is a metabolic disease mainly caused by the production of excess uric acid or uric acid excretion disorder in the body [2]. Excessive uric acid may lead to the sedimentation of urate crystals in joints and gout. Studies had demonstrated that gout is associated with hypertension, atherosclerosis, insulin resistance, cardiovascular diseases and renal diseases [3]. Xanthine oxidase (XO, EC 1.17.3.2) is a key enzyme involved in purine metabolism and biosynthesis of uric acid [4]. XO is a homodimer with a molecular mass of approximately 290 kDa [5]. Each monomer of XO contains one N-terminal domain (20 kDa) including two iron–sulfur centers (Fe/S I and Fe/S II), one central flavin adenine dinucleotide domain (40 kDa), and one C-terminal molybdopterin-binding domain (85 kDa) with the four redox centers aligned in an almost linear fashion [6]. Catalytic reaction occurs at molybdopterin-binding domain [7]. Due to the fact that XO could catalyze the oxidation of hypoxanthine to xanthine and xanthine to uric acid, the inhibition of XO is a main approach for decreasing the production of uric acid and alleviating the progress of hyperuricemia. The general treatment strategy for hyperuricemia was using traditional drugs such as allopurinol and febuxostat. This could effectively inhibit XO activity and alleviate pain caused by gout in the short term, but most of them were found in clinical trials to present adverse side effects [8,9]. Thus, it is critical to develop a natural and efficient XO inhibitory agent. In recent years, the focus of the researches has switched to discovering food-derived bioactive peptides with high XO inhibitory properties and minimal side effects. For example, protein-based peptides such as milk-derived peptide (YLDNY) and walnut protein-derived peptides (WPPKN and ADIYTE) had shown XO inhibitory activity [10,11]. Moreover, several studies had found that marine fish (such as bonito, tuna and shark) protein hydrolysates displayed high XO inhibitory activity [12,13,14,15,16,17].

Round scad (*Decapterus maruadsi*) is a kind of marine fish belonging to the family of mackerel, which is widely found in distributed warm, nearshore waters of China [18]. The current utilization of the round scad is limited due to its dark color, small size, susceptibility to oxidation, and poor flavor [19]. Therefore, this fish is usually considered as a bycatch species and processed into low-value products such as fish meal, fertilizer or even discarded, entailing considerable waste [20]. It is urgently needed to develop higher value-added products to increase the utilization and economic value of the round scad. Recently, the deep processing of round scad has attracted increasing attentions, and some bioactive peptides (antioxidant peptides and memory improving peptides) have been obtained from round scad protein [20,21,22,23]. However, little information is available on the XO inhibitory activity of the peptides derived from round scad hydrolysates (RSHs).

In the present study, the XO inhibitory activity of RSHs prepared by different proteases was investigated. The XO inhibitory peptides in the hydrolysates were further isolated and identified by immobilized metal affinity chromatography (IMAC) and nano-high-performance liquid chromatography mass/mass spectrometry (nano-HPLC-MS/MS). Finally, the identified peptides were synthesized for determining their XO inhibitory activity. The inhibition kinetics study and molecular docking simulation were used for further exploring potential inhibitory mechanism of the XO inhibitory peptides.

## 2. Results and Discussion

### 2.1. XO Inhibitory Activity of RSHs

Since different enzymes have specific cleavage sites on polypeptide chains, three proteases (neutrase, protamex and alcalase) were employed to hydrolyze the round scad proteins. As shown in Figure 1, the XO inhibitory activity and the degree of hydrolysis of the RSHs (obtained by using neutrase, protamex and alcalase) were measured at various hydrolysis times. The hydrolysates obtained by neutrase showed the potent XO inhibitory activity. The XO inhibitory activity of the neutrase hydrolysates was dramatically increased with the increase of hydrolysis time (1–6 h). At 6 h, the XO inhibitory activity reached a maximum value of 62.79 ± 1.41% (15 mg/mL), and the degree of hydrolysis was 15.30 ± 0.16%. Subsequently, the XO inhibitory activity showed a small decrease when the hydrolysis time was over 6 h, which indicated that excessive hydrolysis was not conducive to the hydrolysate to exhibit high XO inhibitory activity. This result showed that both the enzyme type and the time of hydrolysis could impact XO inhibitory activity of RSHs. Li et al. (2018) reported that bonito hydrolysates produced using papain exhibited the highest XO inhibitory ability, compared to walnut hydrolysates and soybean hydrolysates [12]. He et al. (2019) investigated that the tuna protein hydrolysates produced by using alcalase showed XO inhibitory activity [13].

### 2.2. Molecular Weight Distribution of RSHs

It has been reported that peptides with low MW usually could be more easily absorbed and strongly influenced their biological activities [24]. In order to investigate the relationship between MW distribution and XO inhibitory activity, the MW distributions of RSHs obtained with neutrase are shown in Figure 2A. The relative content of fractions with different MW in RSHs was calculated by the percentage of the integral area of the curve (Figure 2B). With the extension of the hydrolysis time, the content of the fraction with a MW below 500 Da significantly increased from 54.60% to 76.33%, while the content of the fraction with a MW range of 1000–3000 Da significantly decreased from 15.53% to 5.00%. In the previous section, we found that as the hydrolysis time was increased from 2–6 h, the XO inhibitory activity of the hydrolysates was dramatically increased, but there was a decrease occurred when the hydrolysis time was over 6 h (Figure 1). This was suggested that low MW (not too small MW) peptides were likely the important contributors of the significant XO inhibitory activity of RSHs. Currently, many low MW peptides with XO inhibitory activity have been identified, such as WML (448.21 Da), PGACSN (547.30 Da), WPPKN (640.74 Da) and ADIYTE (710.73 Da) [11,12]. Therefore, the RSHs (6 h hydrolysates obtained by neutrase) were selected for further separation and purification.

### 2.3. Purification and Identification of XO Inhibitory Peptides

Immobilized metal affinity chromatography (IMAC) was widely used in the separation and purification of metal chelating peptides due to its specific metal biosorption capacity [25]. It has been reported that IMAC can be used as an effective approach to separate low MW peptides because it can reduce the adsorption interference of large proteins [26]. RSHs with the highest XO inhibitory activity (6 h hydrolysates obtained by neutrase) were further purified via IMAC. As shown in Figure 3, two fractions (F1 and F2) were separated from the RSHs. The fraction F2 exhibited higher XO inhibitory activity (IC_50_ = 8.03 ± 0.05 mg/mL) than that of the fraction F1 (IC_50_ = 9.57 ± 0.23 mg/mL). The amino acid sequences of the XO inhibitory peptides in the fraction F2 were further determined by nano-HPLC-MS/MS.

As shown in Figure 4, four XO inhibitory peptides were identified as KGFP (447.53 Da), FPSV (448.51 Da), FPFP (506.59 Da) and WPDGR (629.66 Da). It was found that the identified peptides contained at least one aromatic amino acid, such as F (Phe) and W (Trp). Subsequently, according to the amino acid sequences, the four identified peptides were chemically synthesized for evaluating its XO inhibitory activity. The XO inhibitory activity of the peptides are presented in Table 1. At the same concentration (5 mM), of all identified peptides, tetrapeptide FPSV had the highest XO inhibitory capacity (22.61 ± 1.81%), followed by FPFP (20.09 ± 2.41%), WPDGR (16.21 ± 0.78%) and KGFP (5.43 ± 0.20%). The XO inhibitory capacity of FPSV (22.61 ± 1.81%, 2.24 mg/mL) or FPFP (20.09 ± 2.41%, 2.53 mg/mL) was much higher than that of the isolated fraction F2 (14.14 ± 0.45%, 2.50 mg/mL). All identified peptides were low MW oligopeptides (<650 Da). The low MW peptides seem to exhibit higher XO inhibitory activity and usually could be more easily absorbed than the high MW peptides [11]. Previously, it has been reported that Phe/Trp-containing peptides can be used as good XO inhibitors [10,13,27,28]. Interestingly, the peptides with higher XO inhibitory activity (FPSV and FPFP) contained both Phe and Pro residues. The XO inhibitory activity of peptides seems to be not as strong as some traditional drugs (such as allopurinol, IC_50_ = 6.79 ± 0.25 μg/mL or 0.0499 ± 0.0013 mM in Figure 3), but as natural agents derived from food protein, peptides have less side effects on human health.

### 2.4. Fluorescence Quenching Studies of XO by FPSV and FPFP

Fluorescence chromatographic analysis was used to further explore the binding mechanism between the peptides and XO. There are three types of intrinsic fluorophores in XO, including Trp, Tyr and Phe. As Phe has a low fluorescence quantum yield and the fluorescence of Tyr is almost completely quenched near an amino group, a carboxyl group or a Trp residue, the intrinsic fluorescence intensity of XO is mainly attributed to Trp residues and slightly attributed to Tyr residues [7]. The fluorescence spectra of XO in the absence and presence of FPSV and FPFP under the excitation at 280 nm are shown in Figure 5. It was found that XO had two strong fluorescence emissions at 336 nm and 404 nm wavelengths. Since FPSV and FPFP had no Trp/Tyr residue, they almost had no intrinsic fluorescence under the experimental conditions. With increasing concentration of peptides FPSV (Figure 5A) and FPFP (Figure 5B), the intensity of the peak at 336 nm and 404 nm decreased but without obvious shift. This phenomenon indicated that peptides FPSV and FPFP could directly interact with XO and quench its inherent fluorescence.

The Stern-Volmer plots of FPSV and FPFP are shown in Figure 5C, and the calculated quenching constants K_sv_ and K_q_ at their corresponding temperatures are shown in Table 2. The calculated K_sv_ and K_q_ values of FPSV were 0.0542 × 10^4^ M^−1^ and 0.0542 × 10^12^ M^−1^ s^−1^, respectively. For the peptide FPFP, the K_sv_ and K_q_ values were calculated to be 0.5608 × 10^4^ M^−1^ and 0.5608 × 10^12^ M^−1^ s^−1^, respectively. The values of K_q_ were markedly larger than the threshold of scattering collisional quenching constant (2.0 × 10^−10^ M^−1^ s^−1^) for a biomolecule [29]. This result suggested that the fluorescence quenching mechanism of XO by FPSV and FPFP might be a static rather than dynamic quenching procedure [30].

### 2.5. Reversibility and the Type of Inhibition

The reversibility and the type of XO inhibition by peptides were investigated to further evaluate the inhibitory reaction. The reversibility of FPSV (Figure 6A) and FPFP (Figure 6B) inhibition was assessed by evaluating the initial rate of the XO promoted reaction versus enzyme concentration at different inhibitor concentrations. A family of straight lines all passed through the origin. The slope of the line decreased with increasing inhibitor concentration, indicating that the inhibition of FPSV and FPFP on XO was reversible. This suggested that tetrapeptides could reversibly inhibit the XO activity rather than reducing the effective amount of the enzyme [31], and this implied that interaction between the peptide and XO is mainly a non-covalent intermolecular interaction [29].

The XO inhibitory types of FPSV and FPFP were determined from Lineweaver-Burk double reciprocal plots. The initial rates at various concentrations of substrate (xanthine) with and without FPSV and FPFP were studied. The Lineweaver and Burk plots with and without FPSV and FPFP are illustrated in Figure 7A,B. The Lineweaver-Burk plots of 1/v versus 1/[Xanthine] gave straight lines with different slopes but they intersected one another in the second quadrant, the *K*m (obtained from horizontal axis intercept values) increased and the *V*max (obtained from vertical axis intercept values) decreased, which indicated that FPSV and FPFP were mixed inhibitor [32]. Although this mechanism of inhibition remains to be determined, the finding of mixed-type inhibition of XO suggests that FPSV and FPFP may interact with the amino acid residue of XO domains distal to the substrate binding site, and then attenuate the activity of XO [33].

### 2.6. Molecular Docking and Visual Analysis

Molecular docking and visual analysis were used to predict and understand the interaction between the ligand (peptide) and biomacromolecule (e.g., XO). XO is a molybdenum-containing enzyme that catalyzes the synthesis of uric acid. The molybdenum domain is the key active site in XO, which is embraced by various amino acid residues (such as Phe^649^, Phe^914^, Phe^1009^, Phe^1013^, Asn^768^, Lys^771^, Glu^802^, Leu^873^, Leu^648^, Ser^876^, Arg^880^, Met^770^, Thr^803^, Thr^1010^, Val^1011^, Leu^1014^, Glu^1261^, etc.) [34], among which Glu^802^, Arg^880^ and Glu^1261^ are the key residues that activate the active domain and play a pivotal role in the catalytic reaction [35]. Furthermore, Phe^914^ residues located next to the surface of molybdenum active center also played an important role in the XO inhibitory activity [36].

As shown in Figure 8A, the Lys residue of tetrapeptide KGFP bounded with the Ala^1079^ (2.47 Å), Ser^1080^ (2.65 Å) and Glu^802^ (2.94 Å) of XO by hydrogen bonds, and the Phe residue connect with Phe^798^ (4.24 Å) of XO by π-π stacking. The Phe residue of tetrapeptide FPSV had π-π stacking with Phe^914^ (4.11 Å) and Phe^1009^ (4.74 Å) of XO (Figure 8B). Additionally, the Phe residues in the tetrapeptide FPFP were involved in the π-π stacking with Phe^914^ (4.29 Å), Phe^1009^ (4.89 Å) and Phe^775^ (4.97 Å) of XO (Figure 8C). Meanwhile, the Trp residue of pentapeptide WPDGR was found to have π-π stacking with Phe^914^ (3.91 Å, 4.40 Å), and Phe^1009^ (4.83 Å, 4.95 Å) of XO (Figure 8D). Although all four peptides interact with the pivotal amino acid residues of XO, they had different inhibitory effects on XO. He et al. (2019) found that the XO inhibitory activity was closely associated with the Phe residue on peptides [13]. Besides, it was reported that a bioactive flavonoid from *Spilanthes calva* D.C. exhibiting the potent XO inhibitory activity could be attributed to the formation of π-π stacking between the inhibitor (flavonoid) and Phe^914^ of XO [37]. The Phe residue contained in the peptides (FPSV and FPFP) with potent XO inhibitory activity was found to have π-π stacking with Phe^914^ in the XO. The XO inhibitory activity of the WPDGR was lower than that of the FPSV and FPFP, which might be attributed to the absence of Phe residue and the formation of π-π stacking between Trp (but not Phe) residue and Phe^914^ in the XO. For the KGFP, it contained Phe residue but this residue did not interact with the pivotal amino acid residue (Phe^914^) of XO, which could result in the lower inhibitory effect on XO. The molecular docking results suggested that the π-π stacking between the Phe residue in the peptide and Phe^914^ in the XO was apparently the key influencing factor that accounted for the different inhibitory effects of the four peptides.

## 3. Materials and Methods

### 3.1. Materials

Round scad was purchased from the Vanguard Supermarket (Guangzhou, China). Neutrase, protamex and alcalase were purchased from Cool Chemical Technology Co., Ltd. (Beijing, China). Cytochrome C (12,400 Da), aprotinin (6511.44 Da), bacitracin (1422.69 Da), L-glutathione oxidized (612.63 Da) and glutathione reduced (307.32 Da) were obtained from Macklin Biochemical Technology Co., Ltd. (Beijing, China). Xanthine, xanthine oxidase and chromatographic grade acetonitrile were purchased from Sigma Chemical Co. (St. Louis, MO, USA). All other chemicals and reagents were of analytical grade and purchased from Beiluo Biological Technology Co., Ltd. (Beijing, China).

### 3.2. Preparation of Round Scad Hydrolysates (RSHs)

RSHs were prepared based on the method of Jiang et al. (2014) [21] with slight modifications. The round scad meat (100 g) was minced and then mixed with distilled water (200 mL). The mixtures were hydrolyzed by three proteases at their respective optimum conditions: neutrase (pH 7.0, 50 °C), protamex (pH 7.5, 50 °C) and alcalase (pH 8.0, 50 °C) respectively, in an enzyme to substrate ratio of 0.3% (*w*/*w*) with a reaction time of 2–6 h. Afterwards, the enzyme reaction was terminated by heating the solution in boiling water for 15 min and then centrifuged at 8000× *g* for 20 min at 4 °C. The resulting supernatants were collected and freeze-dried to obtain RSHs and stored at −20 °C until use.

### 3.3. XO Inhibitory Activity Assay

The XO inhibitory activity assay was determined following the method of Masuoka et al. (2015) [38] with some modification. The XO inhibitory activity was measured by quantifying the rate of uric acid produced in the enzyme-catalyzed reaction. All samples were dissolved in 20 mM sodium carbonate buffer at pH 7.5. Briefly, 50 µL of a test sample solution (15 mg/mL) or buffer solution for the blank, and 50 µL of XO solution (0.05 U/mL) were added to the 96-well enzyme linked immunosorbent assay plate for pre-incubation at 37 °C for 30 min. Subsequently, the enzyme reaction was started by the addition of 150 µL of xanthine solution (0.42 mM). During the reaction, the dynamic changes of the absorbance of the sample were continuously recorded at 290 nm. Each sample was tested three times. The XO inhibitory activity was calculated using the following formula:Inhibition (%) = (V_1_ − V_2_)/V_1_ × 100%(1)
where V_1_ is the initial enzymatic reaction rate of the blank group, V_2_ is the initial enzymatic reaction rate of sample group.

### 3.4. Determination of the Degree of Hydrolysis

Degree of hydrolysis was measured according to the method of Wu et al. (2019) [39] with some modifications. Kjeldahl method was used to determine total nitrogen (TPN), and potentiometric titration was used to determine the content of amino nitrogen (AN). Enzymatic hydrolysate (5 mL) was added to 100 mL deionized water and was titrated with 0.1 M NaOH to a pH value of 8.2. Then, an additional 5 mL of formaldehyde aqueous solution and was fully mixed and titrated with 0.1 M NaOH to a pH value of 9.2. The volume (mL) of 0.1 M NaOH consumed after adding formaldehyde was recorded as V_1_. Deionized water was used instead of the sample for blank experiment, and the volume (mL) of 0.1 M NaOH consumed was recorded as V_0_. The calculation formula of AN was as follows: AN (g/mL) = (V_1_ − V_0_) × C × 0.014/5, where C is the actual concentration of NaOH (M), 0.014 is the mass (g) of nitrogen equivalent to 1 mL of 1 M NaOH, and 5 is the volume (mL) of sample. TPN was determined by Kjeldahl method.
Degree of hydrolysis (%) = AN/TPN × 100%(2)

### 3.5. Molecular Weight Distribution of RSHs

The molecular weight (MW) distribution of RSHs was determined by using LC-20AD high-performance liquid chromatography (HPLC) system (Shimadzu Co., Tokyo, Japan) with a TSK-Gel G2000SWXL column (300 × 7.8 mm, 5 µm) according to the method of Zhang et al. (2021) [40] with slight modifications. RSHs (2 mg/mL) were prepared with ultrapure water. Mobile phase A and mobile phase B were acetonitrile with 0.1% (*v*/*v*) Trifluoroacetic acid (TFA) and ultrapure water with 0.1% (*v*/*v*) TFA in isocratic elution program (20% A:80% B). The flow rate was set at 0.5 mL/min. The absorbance was measured at 214 nm. The standard curves of molecular weight were obtained using five standards: cytochrome C (12,400 Da), Aprotinin (6511.44 Da), bacitracin (1422.69 Da), L-Glutathione oxidized (612.63 Da), and Glutathione reduced (307.32 Da). The log of the MW (Da) of the standards was plotted as the function of retention time to get the standard curve: log MW = −0.2212 t + 6.8856 (R^2^ = 0.9964).

### 3.6. Purification of RSHs with Immobilized Metal Affinity Chromatography (IMAC)

IMAC is a good purification technique for proteins and peptides that exhibits the advantages of high selectivity, high binding capacity and high recovery [25]. The RSHs were purified according to the method of Lv et al. (2009) [41] with some modifications. A column (16 mm × 30 cm) was packed with IMAC-Sepharose TM6 Fast Flow (GE Healthcare, Waukesha, WI, USA) and charged with Fe^2+^ (5 bed volumes, 200 mM FeCl_2_ · 4H_2_O). RSHs (20 mg/mL) was filtered through filter membrane (0.22 µm). The nonspecific bound iron was removed with milli-Q water (5 bed volumes), subsequently, 2 mL of the sample solution was loaded onto the column. RSHs were isolated through a two-step elution program. Peptides without affinity to immobilized iron in the column were eluted with distilled water. Then, the bound peptides were eluted using 20 mM phosphate-buffered saline, pH 7.4 (containing 500 mM NaCl), with a flow rate of 2 mL/min. The absorbance of eluates was monitored at 220 nm with UV detector. The elution fractions were collected at room temperature and lyophilized, then stored at −20 °C for further XO inhibitory activity evaluation.

### 3.7. Identification and Synthesis of Peptide

The purified fraction F2 exhibited the strongest XO inhibitory activity was identified by nano-HPLC-MS/MS on a Q Exactive Plus mass spectrometry (Thermo Fisher Scientific, Waltham, MA, USA). A volume of 9 µL of sample was loaded into a chromatographic analytical column (Acclaim PepMap C18, 75 µm × 15 cm) at a flow rate of 300 nL/min. The elution conditions were as follows: mobile phase A (0.1% formic acid in water); mobile phase B (0.1% formic acid in ACN); column temperature (40 °C); linear elution program (0–30 min, 8–30% B). Mass spectra were recorded over an m/z range of 100–3000. The electrospray voltage was 2 kV. Tandem mass spectra were processed by PEAKS Studio version X+ (Bioinformatics Solutions Inc., Waterloo, ON, Canada).

The identified peptide was synthesized via the solid-phase peptide synthesis technique by GL Biochemistry Co., Ltd. (Shanghai, China) for the analysis of its XO inhibitory activity. The synthetic substances were performed through HPLC with a Sinochrom ODS-BP column (250 × 4.6 mm, 5 µm, Shimadzu Corp. Cat, Kyoto, Japan). Then, 5 μL solution was injected to the column and eluted by the system at a flow rate of 1 mL/min. The solvent A and solvent B were 0.1% TFA in 100% acetonitrile and 0.1% TFA in 100% water, respectively. A gradient elution program was used as follows: 0–25 min, 26–51% A; 25.1–30 min, 100% A at a flow rate of 1 mL/min. A UV detector was used to record at 220 nm. Lastly, these identified peptides were verified via mass analysis using electrospray ionization-MS (Shimadzu, Kyoto, Japan), and the purity of the synthetic peptides was over 97%. The experimental conditions of the mass spectrometer were as follows: ion spray voltage, 4.5 kV; nebulizer gas flow, 1.5 L/min; detector voltage, 1.5 kV; de-solvent tube voltage, −20 V; de-solvent tube temperature, 250 °C; block temperature, 200 °C.

### 3.8. Fluorescence Spectroscopy Assay

Fluorescence spectra were measured on an F-7000 Spectrofluorimeter (Hitachi Ltd., Ibaraki, Japan) with a 1.0 cm quartz cell and a thermostatic bath. Briefly, 2 mL XO (0.05 U/mL) was titrated by continuous addition of peptides solution (to final concentrations ranging from 0 to 20.0 × 10^−4^ M). Then, the obtained mixture was properly mixed and equilibrated for 5 min. The slit widths for excitation and emission were set to 2.5 nm. The fluorescence emission spectra were then measured in the emission wavelength range from 300 to 500 nm, under 280 nm excitation wavelengths. The following Stern-Volmer equation was used to analyze the fluorescence quenching [7]:*F*_0_/*F* = 1 + *K*_sv_ [C] = 1 + *K*_q_ *τ*_0_ [C](3)
where *F*_0_ and *F* represent the steady-state fluorescence intensities in the absence and presence of quencher, respectively. *K*_sv_ is the Stern-Volmer quenching constant. [C] is the quencher concentration. *K*_q_ is the quenching rate constant of biomolecule. *K*_q_ = *K*_sv_/*τ*_0_, *τ*_0_ is the average lifetime of biomolecule in the absence of quencher and its value is approximately 10^−8^ s. *K*_sv_ used to be estimated by linear regression plot slope of *F*_0_/*F* versus [27].

### 3.9. Determination of Reversibility and the Type of Inhibition

The plots of ν (ΔOD_290nm_/min) versus the XO (U/mL) at different concentrations of the inhibitors (peptides) were constructed to determine the reversibility of inhibitory behavior on XO [29]. All samples were dissolved in 20 mM sodium carbonate buffer (pH 7.5). Inhibitor solution (0, 5, 10 and 20 mM) and XO solution (from 0 to 0.1 U/mL) were mixed and incubated at 37 °C for 30 min, and the concentration of the substrate xanthine was kept constant at 0.4 mM. The absorbance at 290 nm was then determined. ν is the initial enzymatic reaction rate of the reaction, which was investigated using the same method for the XO inhibitory activity assay.

The type of the inhibition of XO activity by the inhibitors (peptides) was investigated using Lineweaver and Burk kinetic analysis. It was performed via evaluating the initial rate of reaction in the presence and absence of inhibitors at different xanthine concentration. Briefly, the peptide (0 and 5 mM) and XO (0.05 U/mL) were mixed and incubated in the buffer (20 mM sodium carbonate buffer, pH 7.5) for 30 min at 37 °C. Subsequently, different xanthine concentrations (0, 0.05, 0.1, 0.2 and 0.4 mM) were successively added into the solutions. Then, the dynamic changes of the absorbance at 290 nm were recorded, and ν was calculated using the same method above. *K*_m_ and *V*_max_ values, as corresponding kinetic parameters, were obtained from the Lineweaver and Burk double reciprocal plots [31].

### 3.10. Molecular Docking and Interaction Analysis

In order to explore the inhibitory effect of peptides on XO, molecular docking studies were used to explore the potential binding sites and binding affinities between XO and peptides. The molecular docking studies were operated by Discovery studio 2016. The 3D structures of peptides were generated using Chem 3D. The X-ray crystal structure of bovine XO complex with inhibitor TEI-6720 (PDB ID: 1N5X) was downloaded from the Protein Data Bank website (http://www.pdb.org/pdb/home/home.do, accessed on 30 September 2020). The process was performed according to Li et al. (2019) [36]. In docking simulation, peptides were used as ligands while XO was used as a receptor. PyMOL 2.0.2 was used to manage all the 3D structure picture files of biomacromolecules (XO) and ligands (recognized peptides). All water molecules and other small molecules in XO crystal structure were excluded, and polar hydrogen atoms and Gasteiger charges were added. After setting the default parameters, the reverse docking simulation between XO and peptide was performed. The results of molecular docking were further analysed by Discovery studio 2016.

### 3.11. Statistical Analysis

Duncan’s *t*-tests were performed using one-way analysis of variance (ANOVA) on the IBM SPSS statistics 25.0 software. Differences were considered statistically significant at *p* < 0.05.

## 4. Conclusions

In the present study, four novel XO inhibitory peptides (KGFP, FPSV, FPFP and WPDGR) were purified and identified from round scad *(Decapterus maruadsi)* protein hydrolysates (6-h hydrolysis by neutrase). The XO inhibitory peptides, especially for FPSV and FPFP, displayed great potential for XO inhibitory activity. Fluorescence spectra assay demonstrated that the fluorescence quenching mechanism of XO by the inhibitors of both FPSV and FPFP was a static quenching procedure. According to the inhibition kinetics study, the inhibition of both FPSV and FPFP was reversible, and the type of inhibition by both FPSV and FPFP was a mixed one. Molecular docking results revealed that the π-π stacking between Phe residue (contained in peptides) and Phe^914^ (contained in the XO structure) played an important role in the XO inhibitory activity of peptides. These results suggested that peptides obtained from marine fish (*Decapterus maruadsi*) exhibited XO inhibitory effect and could be used as potential natural XO inhibitors.

## Figures and Tables

**Figure 1 marinedrugs-19-00538-f001:**
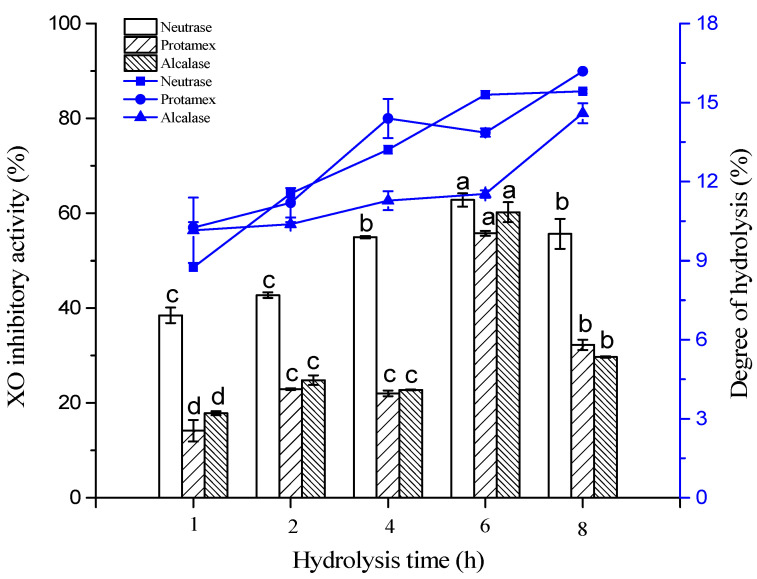
XO inhibitory activity and the degree of hydrolysis of the RSHs obtained by hydrolysis with different proteases. Different letter of ‘a–c’ on the top of same pattern column represents significant difference (*p* < 0.05) in the XO inhibitory activity of RSHs obtained by same protease at different hydrolysis time. For example, the first column represents the XO inhibitory activity of RSHs obtained by neutrase from hydrolysis among 1 to 8 h. There is a significant difference between 6 h (labelled with ‘a’) to both 4 and 8 h (both labelled with ‘b’ which means no significant difference between 4 and 8 h) or to both 1 and 2 h (both labelled with ‘c’). Same rule also used for ‘a–d’ for the RSHs obtained by protamex or by alcalase (middle or last column of different hydrolysis time).

**Figure 2 marinedrugs-19-00538-f002:**
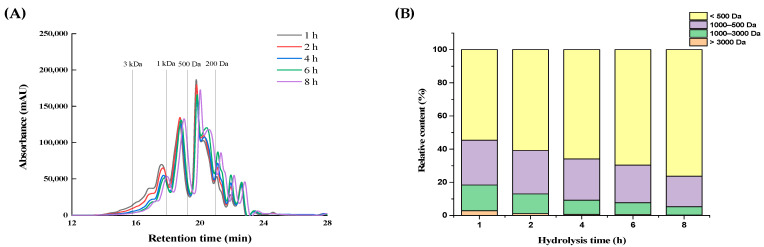
Molecular weight distribution (**A**) of RSHs obtained with neutrase and the relative content (**B**) of fractions with different molecular weight in RSHs.

**Figure 3 marinedrugs-19-00538-f003:**
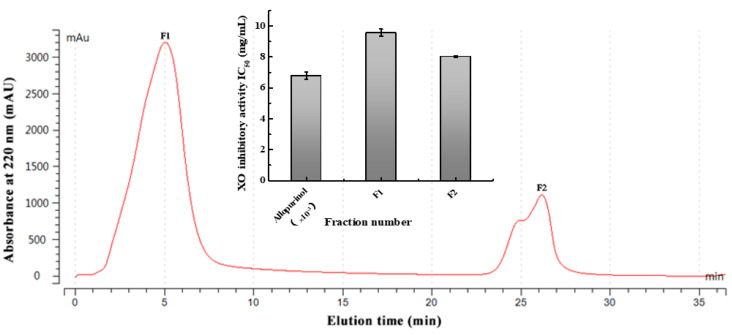
Separation of RSHs by IMAC and the XO inhibitory activity of the peptide fractions.

**Figure 4 marinedrugs-19-00538-f004:**
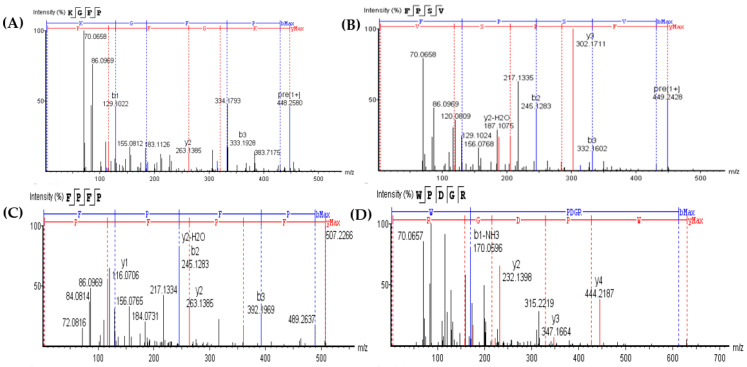
Identification of four XO inhibitory peptides in F2 by nano-HPLC-MS/MS, and the MS/MS spectra of KGFP (**A**), FPSV (**B**), FPFP (**C**) and WPDGR (**D**). The y1–4 and b1–3 are the y-type and b-type fragment ions in the MS/MS spectra.

**Figure 5 marinedrugs-19-00538-f005:**
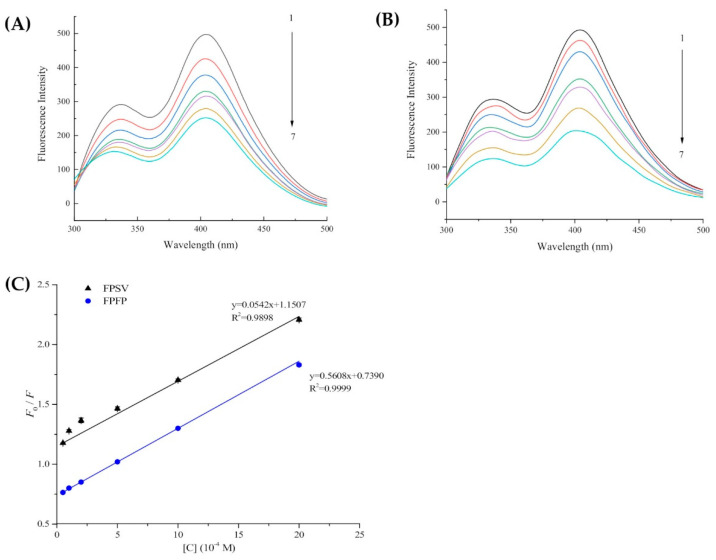
Fluorescence spectra of XO in the presence of FPSV (**A**) and FPFP (**B**) at different peptide concentrations (0, 0.5, 1.0, 2.0, 5.0, 10.0 and 20.0 × 10^−4^ M for curves 1→7, respectively) and the Stern-Volmer plots for the fluorescence quenching of XO by FPSV and FPFP (**C**).

**Figure 6 marinedrugs-19-00538-f006:**
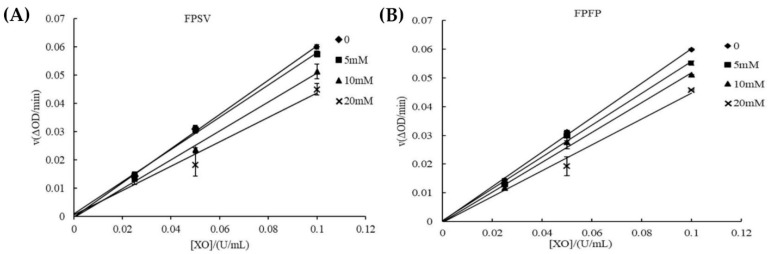
XO inhibition activity of FPSV (**A**) and FPFP (**B**) as a function of enzyme concentration at different inhibitor concentrations.

**Figure 7 marinedrugs-19-00538-f007:**
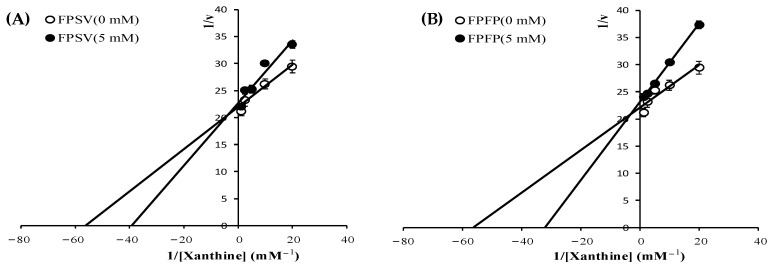
Lineweaver-Burk plots for XO inhibition activity with FPSV (**A**) and FPFP (**B**).

**Figure 8 marinedrugs-19-00538-f008:**
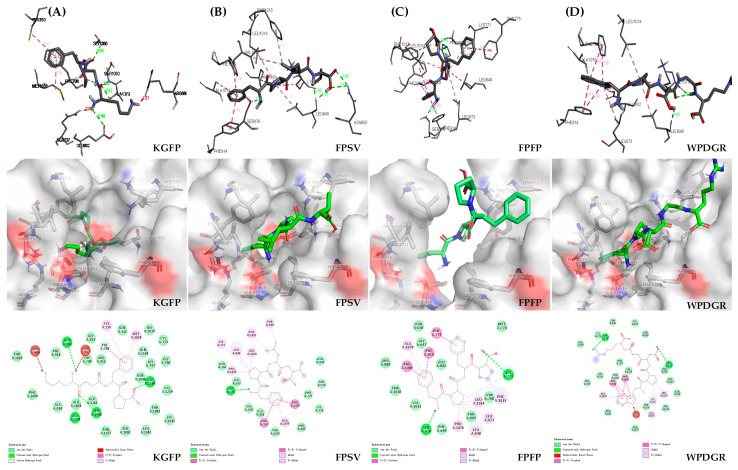
Molecular docking results of KGFP (**A**), FPSV (**B**), FPFP (**C**) and WPDGR (**D**).

**Table 1 marinedrugs-19-00538-t001:** XO inhibitory activity of the identified peptides at the concentration of 5 mM.

Peptides	MW (Da)	Structure Formula	XO Inhibitory Activity (%)
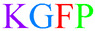	447.53	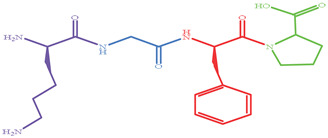	5.43 ± 0.20 ^c^
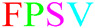	448.51	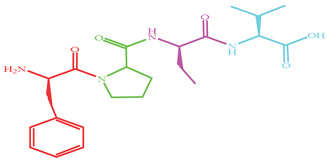	22.61 ± 1.81 ^a^
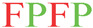	506.59	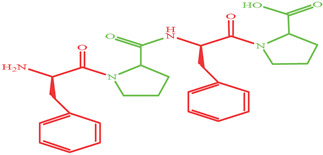	20.09 ± 2.41 ^a^
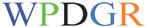	629.66	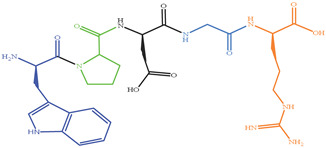	16.21 ± 0.78 ^b^

Values within the same column followed by the same letter are not significantly different (*p* > 0.05).

**Table 2 marinedrugs-19-00538-t002:** Stern-Volmer quenching constants for the interaction of FPSV and FPFP with XO.

Peptides	T (K)	K_sv_ (×10^4^ M^−1^)	K_q_ (×10^12^ M^−1^ s^−1^)	R^2^
FPSV	298	0.0542	0.0542	0.9898
FPFP	298	0.5608	0.5608	0.9999

## Data Availability

Data are contained within the article.

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
