# Peer review of "Purification and Identification of Novel Xanthine Oxidase Inhibitory Peptides Derived from Round Scad (Decapterus maruadsi) Protein Hydrolysates"

_marinedrugs, 2021, doi:10.3390/md19100538_

Round 1

Reviewer 1 Report

The manuscript “Purification and identification of novel xanthine oxidase inhibitory peptides derived from round scad (Decapterus maruadsi) protein hydrolysates” presents a very significant set of results, is very well structured and written in a very clear manner. Its publication is recommended, with some suggestions for minor changes.

Introduction

Page 1, line 45 – I suggest including “is” before “a key enzyme”.

Results and discussion

Page 2, line 81 – I also suggest including “proteins” after “round scad”.

Page 2, line 85 – The inhibitory activity of hydrolysates prepared with Neutrase after 6 hours was not significantly different from those prepared with the other enzymes. Please check this sentence.

Page 2, lines 92 and 94 – I suppose that the dates are not necessary to be indicated.

Page 4, line 129 – Please check these values of IC50.

Page 6, Table 2 – Please check the R2 value of FPSV because it is not in accordance with the value shown in Fig. 5C.

Page 8, line 230 – As mentioned above, the date (2019) is not necessary.

Page 8, line 233 – Please revise the sentence “…very important… inhibitory activity” because it is not clear.

Page 8, Fig. 8 – Unfortunately the numbers of amino acids are too small and it is not possible their identification.

Materials and Methods

Page 9, line 253 – I think that it is “were purchased”.

Page 9, line 275 – Similarly, I suppose that it is “were continuously”.

Page 10, line 298 – Does TFA stands for “trifluoroacetic acid”?

Page 10, line 340 – It is “analyze”.

Page 11, line 357 – I suppose that it is “were used”.

Author Response

Dear Reviewer, 

Many thanks for your constructive revision to improve our research output and manuscript quality. We have tried our best to respond your comments which showed in file for your convenience. 

Much appreciate your speciality knowledge and great effort in helping us all. 

Kind regards

Shaobo Zhou 

Reviewer 2 Report

In this manuscript is described the identification of novel peptides as XO inhibitors. These peptides were purified and identified from round scad hydrolysates. I consider that this is an interesting work; however, some improvements are necessary, in my opinion:

-English should be improved – examples: lines 44/45, 82 (The), 145-146

-In text till line 51 the references are not adequate, given the fact that the text is centered on more general aspects of the subject of the manuscript. For example, adequate reviews should be more suitable. A similar situation occurs in lines 59-63…

-Lines 99-100 “Different letters a-d on the tops of columns indicate significant difference of the sample treated by the same protease at different hydrolysis time (p < 0.05).” – this must be clarified

-In caption of Table, the concentration used of the peptides in the assay must be indicated

-The % of inhibition at 5mM is relatively low. In this context, the authors referred that “The XO inhibitory activity of peptides seems to be not as strong as some traditional drugs, but as natural agents derived from food protein, peptides have less side effects on human health.” I have to agree, but only partially, because to achieve 5mM or higher concentrations in the human body people have to take a very huge quantity… In addition, pharmacokinetics, stability and immunogenicity considerations also must be taken into account. Therefore, I consider that authors must clarify the idea of a potential practical use of these compounds in daily therapeutics.

-The authors referred that the “four identified peptides were chemically synthesized for evaluating its XO inhibitory activity”, however in Experimental it is not clear how they were prepared, purified and identified. In addition, I consider that experimental part should be more developed to allow the reproduction of experiments (example: Determination of Reversibility and Inhibition Type)

Author Response

(The authors gave the same response as above.)

Round 2

Reviewer 2 Report

I read again the manuscript and despite an improvement, some points can still be considered:

-The English still continues needing attention, because authors only changed the few examples of written problems given by the reviewer in the previous review

-in Fig.1, the exact mean of each letter (a, b, and c) must be indicated in the caption

-from a Pharmacist viewpoint, I continue not agreeing with the suggestion of authors that these peptides can be used in treating hyperuricemia because of their low potency of XO inhibition and the corresponding need of very high quantities to achieve a therapeutic effect. Despite this, I consider this an interesting study more because these structures can be a starting point to develop compounds with improved properties, in addition to the valorization of round scad (Decapterus maruadsi)

Author Response

Dear Reviewer, 

Many thanks for your detail comments. We take them carefully consideration. Please see our response in the file. 

Shaobo  
